# Quorum-sensing control of matrix protein production drives fractal wrinkling and interfacial localization of *Vibrio cholerae* pellicles

Boyang Qin [1,2] & Bonnie L. Bassler [1,3] ✉

Bacterial cells at fluid interfaces can self-assemble into collective communities with stunning macroscopic morphologies. Within these soft, living materials, called pellicles, constituent cells gain group-level survival advantages including increased antibiotic resistance. However, the regulatory and structural components that drive pellicle self-patterning are not well defined. Here, using *Vibrio cholerae* as our model system, we report that two sets of matrix proteins and a key quorum-sensing regulator jointly orchestrate the sequential mechanical instabilities underlying pellicle morphogenesis, culminating in fractal patterning. A pair of matrix proteins, RbmC and Bap1, maintain pellicle localization at the interface and prevent self-peeling. A single matrix protein, RbmA, drives a morphogenesis program marked by a cascade of ever finer wrinkles with fractal scaling in wavelength. Artificial expression of *rbmA* restores fractal wrinkling to a Δ*rbmA* mutant and enables precise tuning of fractal dimensions. The quorum-sensing regulatory small RNAs Qrr1-4 first activate matrix synthesis to launch pellicle primary wrinkling and ridge instabilities. Subsequently, via a distinct mechanism, Qrr1-4 suppress fractal wrinkling to promote fine modulation of pellicle morphology. Our results connect cell-cell signaling and architectural components to morphogenic patterning and suggest that manipulation of quorum-sensing regulators or synthetic control of *rbmA* expression could underpin strategies to engineer soft biomaterial morphologies on demand.

A hallmark of soft biological tissues and materials is their ability to fold, bend, and self-organize into complex yet functional shapes. Indeed, across different domains of life, genetically identical cells self-assemble into ordered, spatially heterogenous structures at length scales far larger than that of the individual cells. Consequently, geometric patterns and global architectures facilitate biological functions in multicellular organisms, such as the fractal architecture of the mammalian lung bronchial tree that minimizes metabolic load and entropy production during gas exchange[1], the liquid-crystal ordering of exoskeleton cells in jeweled beetles that give rise to circularly polarized iridescence[2], the development of deep fissures and convolutions in the cerebral cortex that promote efficient cortex packing[3,4], and the hierarchical looping and folding of gut villi that increase surface area for nutrient absorption[5,6].

Bacterial cells also form collectives with striking morphologies on solid surfaces and at liquid interfaces, known as biofilms and pellicles[7,8],

[1]Department of Molecular Biology, Princeton University, Princeton, NJ 08544, USA. [2]Department of Mechanical and Aerospace Engineering, Princeton University, Princeton, NJ 08544, USA. [3]The Howard Hughes Medical Institute, Chevy Chase, MD 20815, USA. ✉e-mail: bbassler@princeton.edu

respectively. Cells in biofilms and pellicles share public goods and are more resilient to antibiotics, immune clearance, and physical stresses than are their individual, isogenic planktonic counterparts[9,10]. As a result, these multicellular communities play important roles in medical infections and industrial biofouling. Single-cell level studies of developing biofilms revealed the emergence of internal cell ordering[11,12], architectural responses to flow perturbations[13], and collective cellular flow[14]. Analogous to the macroscopic structures present in eukaryotic systems, surface associated bacterial biofilms develop striking morphologies including corrugations[15,16], ordered radial stripes[17], herringbone-like wrinkles[18], and delaminated blisters[17].

We recently reported a hierarchical fractal wrinkling program that drives morphogenesis during *Vibrio cholerae* pellicle formation at a fluid–fluid interface[19]. *V. cholerae* is a global pathogen that annually infects ~3 million people worldwide[20]. Germane to the present work, liquid-associated *V. cholerae* multicellular communities serve as aquatic reservoirs of infection[21,22]. Using a custom stereomicroscope setup that adaptively tracked morphological features, we characterized the sequential morphological stages of *V. cholerae* pellicles as they formed at the interface between the growth medium and mineral oil. Four stages of morphogenesis occur during pellicle maturation: growth of interface localized colonies, onset of primary wrinkles, formation of secondary ridge instabilities, and finally, emergence of a cascade of finer structures with fractal-like scaling in wavelength[19]. Our earlier study explored the roles of colony microstructures and mechanical stresses, however, it did not probe the molecular mechanisms nor the matrix components underlying the observed fractal wrinkling morphogenesis.

Here, using mutagenesis, measurements of gene expression, time course imaging, and fractal analyses, we uncover the biocomponents responsible for the *V. cholerae* pellicle fractal wrinkling program. We find that primary wrinkling and secondary ridge instability can occur in the absence of the extracellular matrix protein RbmA. The subsequent fractal wrinkling stage, however, requires RbmA (Fig. 1). Indeed, inducible expression of *rbmA* in a Δ*rbmA* mutant triggers rapid formation of fractal wrinkles, and the level of fine-scale wrinkling that occurs correlates with the strength of *rbmA* expression. Moreover, expression of *rbmA* is sufficient to drive fractal wrinkling in a Δ*rbmA* mutant even after secondary ridges have formed. The matrix proteins

RbmC and Bap1 function redundantly to maintain pellicle stability at the fluid interface. When *rbmC* and *bap1* are deleted, the pellicle peels off of the interface and collapses. Quorum-sensing (QS) regulation of *rbmA*, *rbmC*, and *bap1* expression orchestrates the community-level commitment to fractal wrinkling morphogenesis. *V. cholerae* cells containing a mutation in the central regulator, LuxO, that locks them into the low-cell-density QS regime cannot achieve fractal wrinkling. Deletion of *hapR*, encoding the QS high-cell-density master regulator, also locks the cells into the low-cell-density QS mode. Surprisingly, this change does not eliminate fractal wrinkling. We show that the regulatory small RNAs (sRNAs) Qrr1-4, that function between LuxO and HapR in the QS hierarchy, are responsible for the differing morphologies. Specifically, the Qrr sRNAs regulate the fractal wrinkling program via two opposing mechanisms. First, the Qrr sRNAs launch the initial morphogenic stages via repression of *hapR*, leading to activation of genes encoding matrix components (Fig. 1). Second, the Qrr sRNAs suppress fractal wrinkling via a HapR-independent, VpsR-dependent mechanism (Fig. 1). VpsR is a transcription factor that activates matrix gene expression. Thus, Qrr sRNA repression of *vpsR* lowers production of matrix components. By systematically controlling Qrr sRNA or RbmA matrix protein production, we show that bacterial pellicles can be manipulated "on demand" to form wrinkling morphologies with varying fractal dimensions. Thus, the understanding we garnered here through our molecular analyses could enable precise patterning of soft, living, active biomaterials.

## Results

### The matrix protein RbmA is required for fractal wrinkling of *V. cholerae* pellicles

To explore the molecular mechanism underlying *V. cholerae* pellicle fractal wrinkling, we begin by probing the roles of key matrix proteins. As we have done in the past[14,17,19], we rely on a commonly used hyper-matrix-producing *V. cholerae* strain carrying the *vpvC*[W240R] mutation. The strain is denoted "Rg" due to the rugose colony morphology it displays on agar surfaces. The *vpvC*[W240R] missense mutation increases VpvC diguanylate cyclase activity, which elevates the concentration of the second messenger molecule cyclic diguanylate (c-di-GMP)[23]. In *V. cholerae*, increased c-di-GMP binds to and activates the VpsT[24] and VpsR[25] transcription factors, which in turn, activate expression of genes encoding vibrio polysaccharide (VPS) biosynthesis enzymes and matrix proteins (Fig. 1).

There are three key *V. cholerae* extracellular matrix proteins, RbmA, RbmC, and Bap1. RbmA is located at the cell poles where it binds mother-daughter cells together[26]. RbmA also associates with the VPS[27]. Bap1 adheres cells to the solid surface, and RbmC/Bap1 encase clusters of cells[26] and alter the overall biofilm surface energy/wettability[28]. We start by assessing the role of RbmA in pellicle morphogenesis. The Rg strain underwent the complete fractal wrinkling program (Fig. 2a), as characterized by the emergence of a cascade of fine wrinkles. By contrast, the Rg Δ*rbmA* mutant formed a pellicle and progressed through the primary wrinkling and ridge instabilities stages; however, it failed to achieve the fractal wrinkling (Fig. 2b, Movie S1). Consistent with this trend, on a solid surface, the Rg Δ*rbmA* strain also displayed a reduction in fine-scale corrugations compared to the Rg parent strain (Fig. 2c, d). Segmentation of the connected pellicle wrinkling features of the Rg parent (Fig. 2e) and of the Rg Δ*rbmA* mutant (Fig. 2f) confirms that without RbmA, pellicle morphogenesis stalls at the secondary ridge morphology stage with no sign of formation of higher order wrinkle structures. Quantitative fractal scaling analyses of the connected wrinkle features demonstrate that the Rg strain reaches a fractal dimension of $\delta = 1.4$ at 74 h, while the Rg Δ*rbmA* strain shows no fractal scaling, i.e., $\delta \sim 1$ (Fig. 2g, h). A fractal scaling larger than 1 indicates increased complexity in the surface geometry of the wrinkling features. Thus, this result suggests that RbmA enables *V. cholerae* pellicles to gain additional surface area

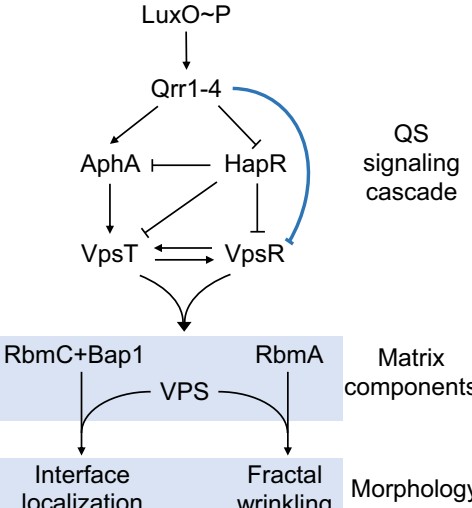

**Fig. 1 | QS controls the production of matrix components and fractal wrinkling during *V. cholerae* pellicle morphogenesis.** A simplified scheme of the *V. cholerae* QS system showing key low-cell-density regulatory interactions and targets. The present work reveals the components required for pellicle interfacial localization and fractal wrinkling (blue box) and the new regulatory connection between the Qrr1-4 sRNAs and VpsR (blue repression symbol). See text for details.

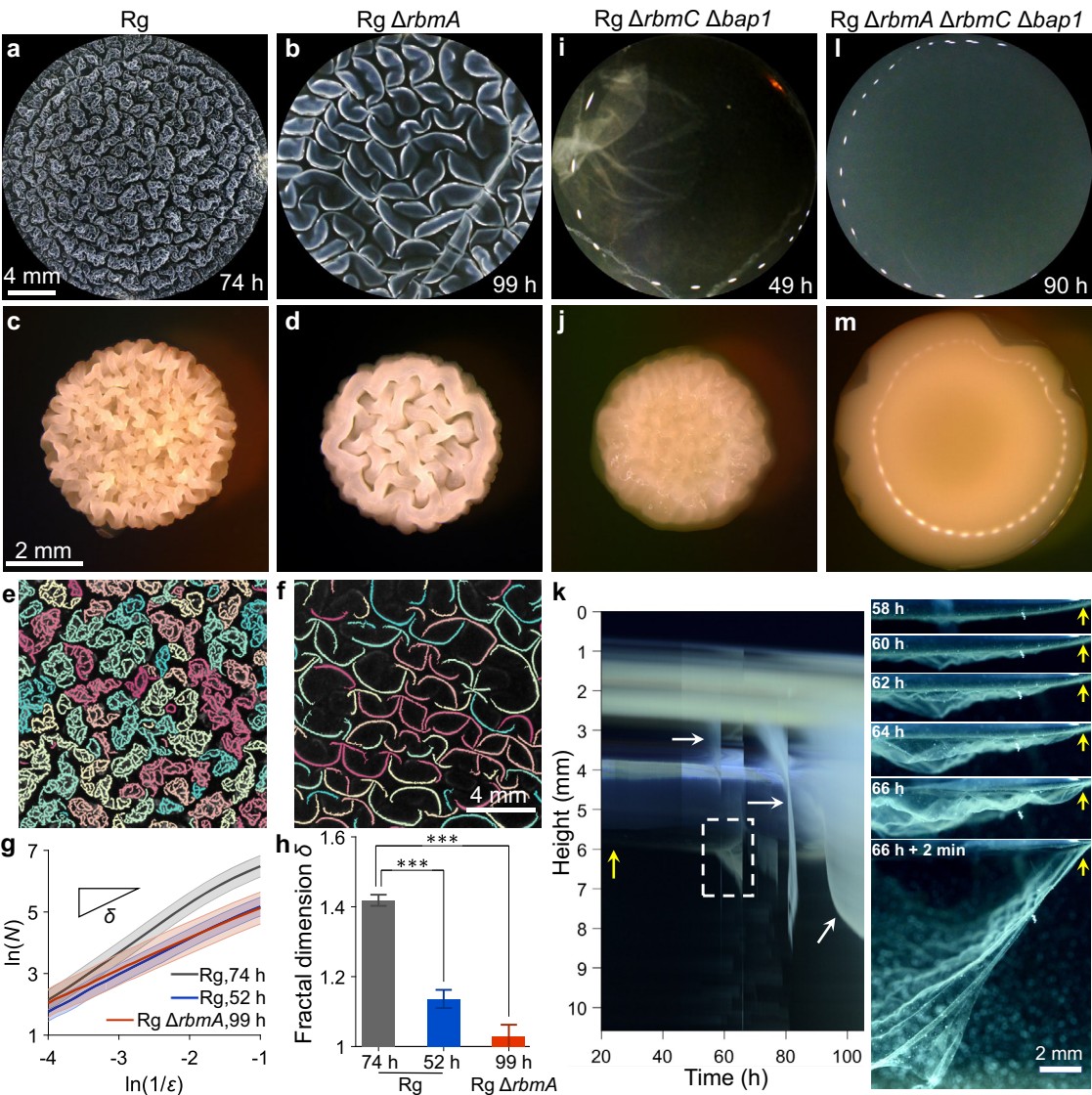

**Fig. 2 | Matrix proteins drive fractal wrinkling and enable interfacial localization during *V. cholerae* pellicle morphogenesis. a, b, i, l** Pellicle morphologies at the liquid–liquid interface and **c, d, j, m** morphologies on solid agar surfaces of the indicated strains. The pellicles were imaged at the designated time points and their morphologies on solid agar surfaces were imaged at 51 h of growth. **e, f** Segmented pellicle wrinkle features for **e** the Rg strain during the fractal wrinkling regime at 74 h and **f** the Rg Δ*rbmA* strain at 99 h, time points showing the fully matured morphologies. The different colors denote disconnected wrinkle features. **g** The Hausdorff dimensions obtained from the scaling exponent of the box count $N$ versus the box length scale $\varepsilon$ from the images of the Rg strain at the ridge instability stage (52 h), fractal stage (74 h, as in **e**) and the Rg Δ*rbmA* strain at long times (99 h,

as in **f**). Shaded areas denote standard deviations from biological replicates ($n = 3$). **h** Fractal dimensions for the indicated strains and time points in **g**. Error bars denote standard deviations from biological replicates ($n = 3$). Unpaired *t*-tests with Welch's correction were performed for statistical analyses. *P* values are: *, <0.05; **, <0.01; ***, <0.001; ****, <0.0001; n.s. (not significant), >0.05; and n.d., not detected. **k** Left: space-time kymograph of pellicle detachment from the interface for the Rg Δ*rbmC* Δ*bap1* strain. The yellow arrow denotes the fluid meniscus in the plane of focus. The white arrows denote detachment events outside of the plane of focus. The dashed box indicates the region shown in the rightmost panel. Right: side-view snapshots leading to the detachment of the pellicle from the meniscus in the plane of focus as indicated in the kymograph.

associated with the fractal nature of the wrinkled structures. This feature may contribute to the capture of nutrients and enhanced cell growth[29].

## The matrix proteins RbmC and Bap1 function redundantly to maintain the *V. cholerae* pellicle interfacial stability

Sustained growth at liquid interfaces is critical for bacterial pellicle communities to access oxygen[30], and potentially facilitate transmission to human hosts as biofilm-like aggregates for pathogenic bacteria[21,31]. In *V. cholerae*, liquid-liquid interfacial localization begins with the formation of individual microcolonies followed by formation of a second layer of microcolonies that fills the meniscus[19]. Subsequent wrinkling mechanical instabilities occur in the confluent layer, but

importantly, the developing pellicle remains attached to the liquid interface throughout morphogenesis. While the *V. cholerae* Rg Δ*rbmC* Δ*bap1* mutant forms an initial confluent layer, it subsequently undergoes a drastically different pellicle program from the Rg parent and the Rg Δ*rbmA* mutant (Fig. 2i, Movie S2). As compressive stress builds up, detachment sites nucleate at the interface and rapidly increase in amplitude, leading to irregular wrinkles that break away from the interface (Fig. 2k, Movie S3). The pellicle subsequently self-peels from the interface and catastrophically collapses over a short time scale (Fig. 2k, i). The interfacial self-peeling and the drastic change in the fate of the community require the loss of both RbmC and Bap1, as the Rg Δ*rbmC* and Rg Δ*bap1* mutants undergo the complete fractal wrinkling morphogenesis program similar to the Rg parent. (Fig. S1a, b).

Consistent with the altered pellicle properties at the liquid interface, on an agar surface, the Rg Δ*rbmC* Δ*bap1* strain possesses dramatically reduced corrugations compared to the Rg strain (Fig. 2j). Hence, RmbC and Bap1 function redundantly to dictate the spatial outcome and the interface localization of the pellicle community.

We wondered if RbmA and Bap1/RbmC function additively in controlling pellicle morphogenesis. To test this possibility, we constructed a triple mutant by deleting *rbmA* from the Rg Δ*rbmC* Δ*bap1* strain. Indeed, all pellicle wrinkling features were completely eliminated (Fig. 2l) as were corrugations on solid surfaces (Fig. 2m). These findings show the synergistic roles of RbmA with that of RbmC/Bap1 in establishing robust communities at the liquid interface. Since the Rg Δ*upsL* mutant which cannot synthesize VPS also does not form pellicles (Fig. S1c), we infer that the protein scaffold formed between the VPS and any of the matrix proteins is essential for the initiation of pellicle morphogenesis. Previous characterization of *V. cholerae* biofilm material properties showed that RbmA is responsible for maintaining elasticity, VPS is crucial for supporting large strains before yielding, and RbmC and Bap1, together, modulate the biofilm surface chemistry[28]. Indeed, these material features are consistent with our current observations.

### *V. cholerae* pellicle fractal wrinkling dimensions correlate with *rbmA* expression levels

Given our finding of the potential overarching role of RbmA in pellicle fractal patterning, we examined whether complementation of *rbmA* in the Rg Δ*rbmA* strain was sufficient to restore the fractal wrinkling program. We engineered arabinose inducible *rbmA* on a plasmid (designated p*rbmA*) and introduced the construct into the Rg Δ*rbmA* mutant. Indeed, addition of the arabinose inducer at the start of the experiment triggered the emergence of fractal wrinkling (Fig. 3a–d). In a titratable manner, the small-wavelength wrinkling features intensified in response to increasing inducer concentration. At 0.002% arabinose (Fig. 3d, Movie S4), the level of fractal wrinkling was maximal, and the pellicle morphology was indistinguishable from that of the Rg strain at identical growth times (Fig. 2a). Fractal analysis of the pellicle wrinkles demonstrated that, as inducer concentration was increased in steps from none to 0.002%, the fractal dimensions of the wrinkling features also increased (Fig. 3i), indicating the emergence of self-similar wrinkles and an increase in shape complexity. Additionally, the power spectra density of the wrinkling features shifted toward smaller wavelengths (Fig. 3j), confirming the emergence of small-sized wrinkling features. To verify that the restoration of fractal wrinkling was due to elevated RbmA, rather than unanticipated changes in other matrix components caused by synthetic modulation of *rbmA* expression, we used qRT-PCR to measure expression of genes encoding other matrix proteins in the Rg, Rg Δ*rbmA*, and Rg Δ*rbmA* p*rbmA* strains under pellicle forming conditions (Fig. 3k). First, we verified that there was no detectable *rbmA* transcript in the Rg Δ*rbmA* mutant that fails to display fractal wrinkling. Second, consistent with the pellicle morphologies, 0.002% arabinose induction restored *rbmA* expression in the Rg Δ*rbmA* p*rbmA* strain to the same level as that in the Rg parent. Crucially, expression of other matrix genes remained unchanged (Fig. 3k). Our control experiment showed that introduction of the empty vector (pEVS) caused no changes in the Rg Δ*rbmA* strain pellicle morphology (Fig. 3g), fractal dimension (Fig. 3i), or matrix gene expression (Fig. 3k). Finally, overexpression of *rbmC* or *bap1* the Rg Δ*rbmA* mutant did not restore fractal wrinkling (Fig. S2). Hence, we conclude that RbmA drives the onset of the fractal wrinkling program, and the wrinkling fractal dimension correlates with the level of RbmA synthesis.

### Overexpression of *rbmA* eliminates formation of large-scale wrinkle structures and ridge instabilities in *V. cholerae* pellicles

As we demonstrated above, RbmA is required for *V. cholerae* pellicles to produce fractal wrinkles, suggesting that excess expression of *rbmA* above that normally expressed in the Rg strain could increase fractal wrinkling further, possibly leading to higher fractal dimensions. We tested this idea using the inducible p*rbmA* construct in the Rg Δ*rbmA* mutant. Surprisingly, as inducer levels were increased above 0.002%, suppression of large-scale structures, including primary wrinkling and secondary ridges, occurred (Fig. 3e, f). Small-scale structures dominated, eliminating the self-similar scaling from large-scale to small-scale structures that is the hallmark of fractal wrinkling. As a result, pellicle fractal dimensions decreased at arabinose levels higher than 0.002% (Fig. 3i). Consistent with this morphological change, the distribution of pellicle feature lengths shifted toward small length-scales at 0.005% inducer level (Fig. 3j). Thus, overexpression of *rbmA* generates small size wrinkles and eliminates the self-similar cascade of wrinkles that span small to large wavelengths. We verified that *rbmA* was overexpressed, as at 0.005% arabinose, *rbmA* transcription was four-times higher than that at 0.002% (Fig. 3k). The reduction in large-scale morphological features was not due to impaired cell growth (Fig. S3). Hence, excess RbmA production beyond that normally present in the Rg strain suppresses fractal wrinkling, i.e., the self-similar scaling of wrinkle wavelengths across a range of length-scales. Rather, fine-scale features are favored at the expense of large-scale features, leading to a distinct morphology. This observation suggests that the Rg strain apparently produces the optimal level of RbmA required to generate fractal wrinkling and to maximize fractal dimension. Since RbmA biosynthesis requires resource expenditure, achieving a high fractal dimension using minimal RbmA could be energetically economical, providing survival advantages at the population level.

### *V. cholerae* pellicle fractal wrinkling can be triggered after secondary ridge instabilities occur

We wondered whether there was a restricted time window in which RbmA could promote pellicle fractal wrinkling, or alternatively, whether RbmA could remodel an already matured pellicle structure. To test these possibilities, we used the Rg Δ*rbmA* mutant carrying the inducible p*rbmA* construct but supplied the arabinose inducer at late times during pellicle development. Addition of 0.002% arabinose at 96 h post-inoculation, the time when secondary ridges begin to emerge (Fig. 4a–c), and at 119 h post-inoculation, the time when secondary ridges have formed (Fig. 4d–f), both triggered the emergence of fractal wrinkling within 20 h. Increased segregation occurred between connected wrinkle features when the inducer was supplied subsequent to secondary ridge formation compared to when inducer was added at the time of inoculation (Fig. 4g, h). Furthermore, individual features were separated by large gaps devoid of wrinkle features. Thus, expression of *rbmA* drove decorations of the large-scale structures that formed in its absence with small-scale wrinkles. To further quantify the dynamics of remodeling, we monitored the time evolution of pellicle fractal dimensions following inducer administration at different times (Fig. 4i). Unlike the minor increase in fractal dimension that occurred when no arabinose was provided, induction with 0.002% arabinose at 72 h (prior to secondary ridge formation), 96 h (onset of secondary ridge formation), and 119 h (post-secondary ridge formation) elicited increases in fractal dimensions within 3 h. The fractal dimensions of the pellicles reached a common plateau value of $\delta = 1.4$ independent of the timing of arabinose supplementation, including immediately after inoculation (0 h, Fig. 3i). This result suggests that the timing of RbmA production provides a path to modulate fractal patterning, whereas the level of RbmA produced dictates the final fractal dimension. Hence, we expect that biological or environmental mechanisms that impinge on *V. cholerae* RbmA synthesis will modify pellicle morphogenesis. Moreover, given that temporally controlled expression of *rbmA* enables rapid production of pellicle fractal morphologies, synthetic strategies that tune the amount and timing of *rbmA* expression could be used to design active materials possessing combined patterns of large wrinkles/ridges and fine fractal wrinkles.

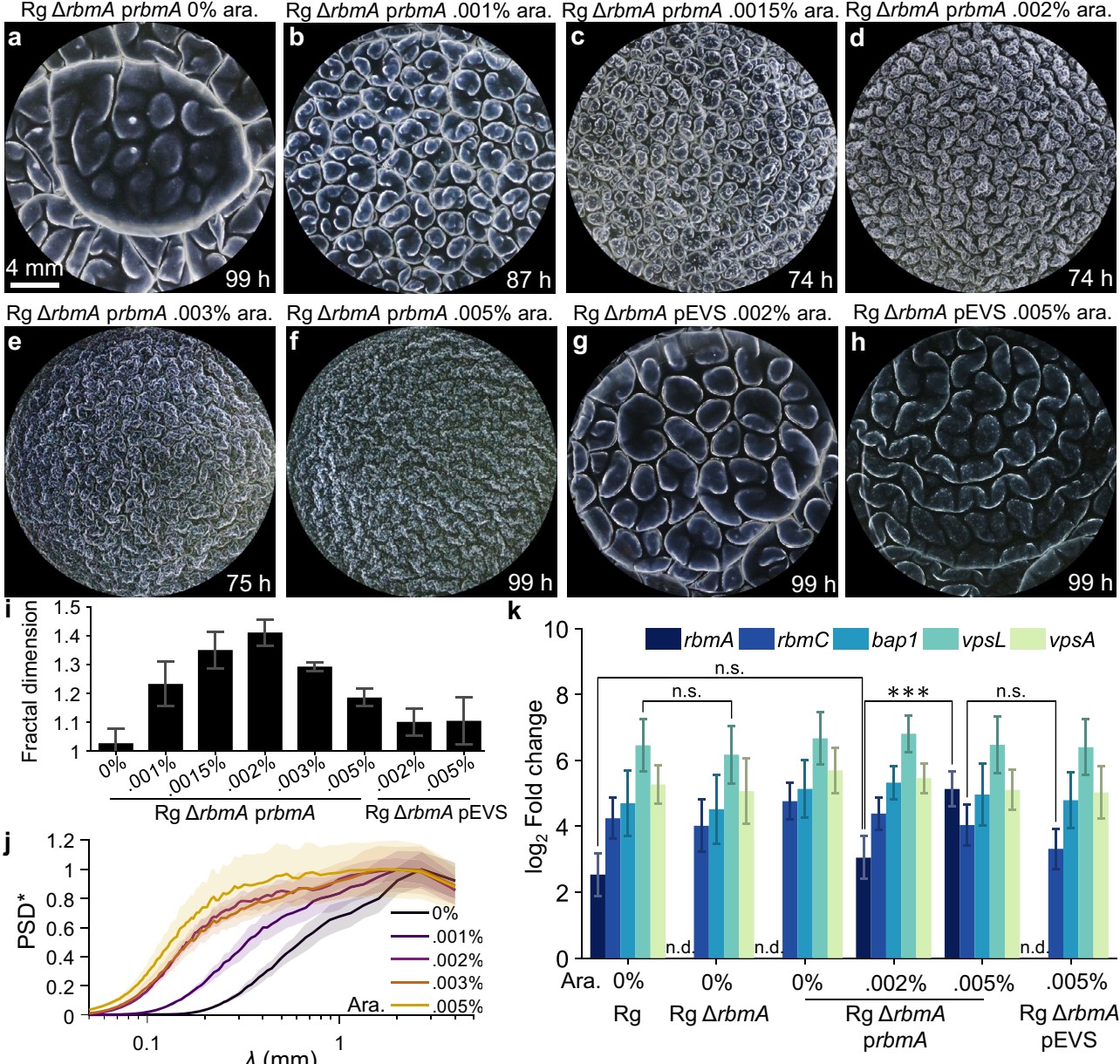

**Fig. 3 | Induction of *rbmA* expression restores fractal wrinkling to *V. cholerae* Δ*rbmA* pellicles. a–f** Pellicle morphologies of the Rg Δ*rbmA* strain expressing *rbmA* from the pEVS143 vector (designated p*rbmA*) at the indicated arabinose inducer concentrations. **g–h** Pellicle morphologies of the Rg Δ*rbmA* strain carrying the empty pEVS143 vector (denoted pEVS) at the designated arabinose inducer concentrations. **i** Fractal dimensions of the wrinkle features for the Rg Δ*rbmA* strain carrying p*rbmA* or the empty pEVS vector at the designated arabinose inducer concentrations and at the time points shown in **a–h**, at which plateau values of fractal dimensions had been reached. The bars correspond to the images in **a–h**. Error bars denote standard deviations of biological replicates ($n = 3$). **j** Distributions

of wrinkling feature wavelength (designated $\lambda$), measured by the normalized power spectra density (designated PSD*) of the wrinkle vesselness images at the designated arabinose inducer concentrations for the Rg Δ*rbmA* strain carrying p*rbmA*. Shaded areas denote standard errors from biological replicates ($n = 3$).
**k** Expression of matrix genes in the designated strains and at the indicated arabinose inducer concentrations assessed by qRT-PCR. Data are presented relative to that from wildtype *V. cholerae*. Error bars denote standard deviations from biological replicates ($n = 3$). *P* values are: *, <0.05; **, <0.01; ***, <0.001; ****, <0.0001; n.s. (not significant), >0.05; and n.d., not detected.

## QS regulates progression through the *V. cholerae* pellicle fractal wrinkling program

QS, the bacterial chemical communication process that controls group behaviors, regulates matrix protein production in *V. cholerae* (Fig. 1). QS relies on the synthesis, detection, and group-wide response to extracellular signal molecules called autoinducers. In *V. cholerae*, information encoded in autoinducers is funneled to the central QS regulator called LuxO[32,33]. The low-cell-density state is the relevant QS mode for pellicle formation: Under this condition, in the absence of autoinducers, LuxO is phosphorylated, and it activates expression of

genes encoding a set of key regulatory small RNAs (sRNAs) called Qrr1-4. Qrr1-4 positively regulate matrix gene expression via two mechanisms. First, the Qrr sRNAs act post-transcriptionally to induce production of the low-cell-density master regulator called AphA. Second, the Qrr sRNAs suppress the production of the high-cell-density master regulator HapR. Both of these Qrr1-4 functions promote matrix biosynthesis because AphA activates while HapR represses matrix gene expression (Fig. 1). QS regulation of matrix biosynthesis is muted by the high levels of c-di-GMP present in the Rg strain. For this reason, in our QS analyses below, we use wildtype *V. cholerae* as the parent strain.

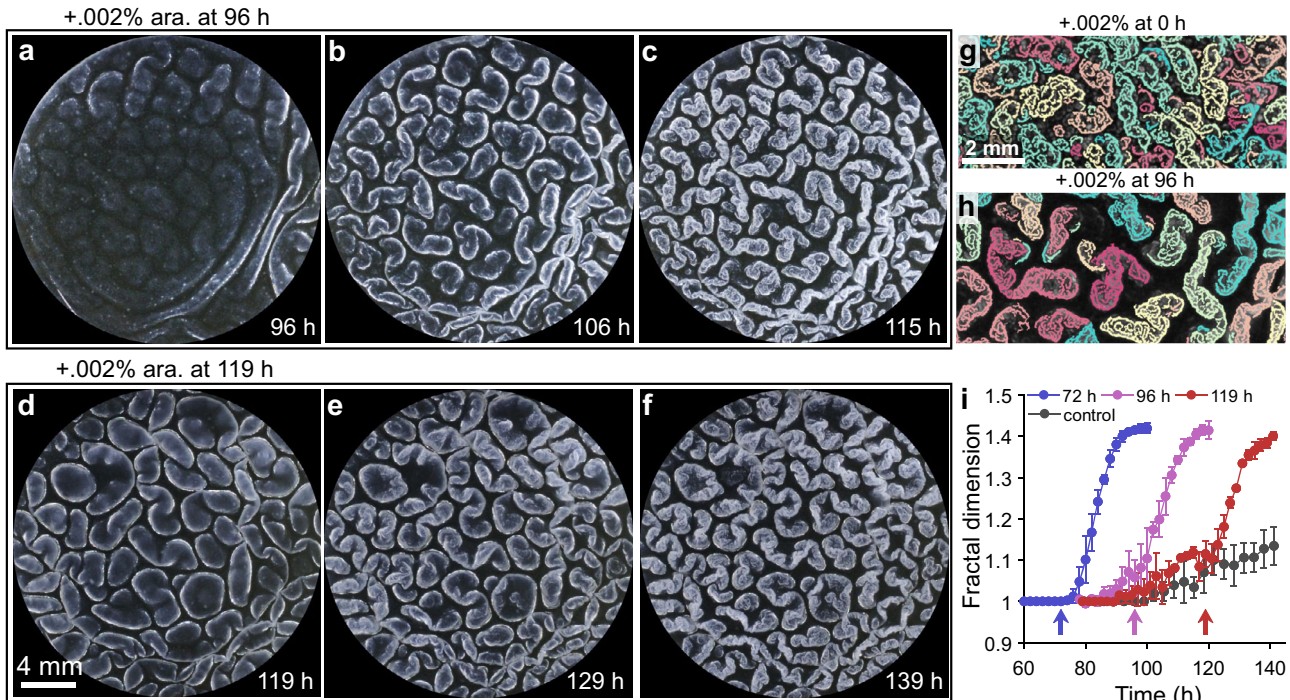

**Fig. 4 | Pellicle fractal wrinkling can be triggered on-demand via induction of** ***rbmA* expression. a–c** Pellicle morphologies of the Rg Δ*rbmA* strain carrying p*rbmA* induced with 0.002% arabinose supplied 96 h after inoculation, when secondary ridges begin to emerge. Images taken at the indicated time points. **d–f** As in **a–c**, with arabinose supplied at 119 h, following secondary ridge formation. Images taken at the indicated time points. **g,h** Segmented pellicle wrinkle features for the Rg Δ*rbmA* strain carrying p*rbmA* to which 0.002% arabinose had been administered at **g** 0 h and imaged at 74 h and **h** 96 h and imaged at 115 h. The different colors denote disconnected wrinkle features. **i** Time development of pellicle wrinkle fractal dimensions of the Rg Δ*rbmA* strain carrying p*rbmA* following induction with 0.002% arabinose at the indicated time points shown by the arrows and the control set without induction. Error bars indicate standard deviations from biological replicates ($n = 3$).

We investigated which *V. cholerae* QS components control the morphological stages leading to fractal wrinkling. Consistent with the known QS regulatory hierarchy, introduction of the *luxO*[D61E] allele, encoding a LuxO phospho-mimetic that locks the cells in the low-cell-density QS state[34] (see Fig. 1), causes wildtype *V. cholerae* to form robust pellicles (Fig. 5a, Movie S5) and an opaque surface morphology on solid agar (Fig. S4a). LuxO[D61E]-directed activation of pellicle formation requires Qrr1-4: Deletion of the *qrr*1-4 genes (Fig. 5b) or deletion of the downstream Qrr1-4 target *aphA* (Fig. S5), completely abrogated pellicle formation. Indeed, quantitation of matrix gene expression in the *luxO*[D61E] Δ*aphA* and the *luxO*[D61E] Δ*qrr*1-4 strains showed large reductions in VPS biosynthesis gene expression compared to that in the *luxO*[D61E] strain (Fig. 5c, *vpsL* and *vpsA* were measured as representative genes for the two VPS biosynthesis clusters[35,36]). This result indicates the critical role the Qrr sRNAs play in activating *aphA* expression, which in turn, initiates matrix biosynthesis at low cell density. Qrr1-4 also repress *hapR*, and HapR directs the high-cell-density QS program[37]. Thus, *hapR* expression was higher in the *luxO*[D61E] Δ*qrr*1-4 strain than in either the *luxO*[D61E] strain or the *luxO*[D61E] Δ*aphA* strain (Fig. 5c). These results suggest that in the absence of Qrr1-4 repression, HapR is unleashed, and it represses matrix gene expression. Hence, the Qrr1-4 sRNAs are the key regulators in the QS hierarchy that enable formation of robust pellicle structures and allow the subsequent progression to primary wrinkling and secondary ridge morphologies.

**The QS Qrr sRNAs suppress *V. cholerae* pellicle fractal wrinkling via a HapR-independent mechanism**

Despite being locked in the low-cell-density state and forming a pellicle, the *V. cholerae luxO*[D61E] strain, to our surprise, failed to achieve fractal wrinkling (Fig. 5a). Specifically, following development of primary wrinkles, pellicle morphology progression stalled at the secondary ridge formation stage. By contrast, the Δ*hapR* mutant, which is also locked in the low-cell-density regime, exhibited fractal wrinkling (Fig. 5d, Movie S6). Likewise, the solid surface-associated morphology of the Δ*hapR* mutant showed a higher level of corrugation than that of the *luxO*[D61E] strain (Fig. S4c). We investigated the mechanism driving these differences by perturbing components in the QS pathway. First, introduction of the Δ*hapR* mutation into the *luxO*[D61E] strain was not sufficient to restore fractal wrinkling (Fig. 5e, Movie S7) or the corrugated solid surface-associated morphology (Fig. S4d). This result implies that a regulator located downstream of LuxO but upstream of HapR must be responsible for the difference. The Qrr sRNAs are the only regulators known to be located between LuxO and HapR in the QS hierarchy (Fig. 1). Indeed, the *luxO*[D61E]Δ*hapR* Δ*qrr*1-4 mutant displays fractal wrinkling (Fig. 5f, Movie S8) and its solid surface-associated morphology is as corrugated (Fig. S4e) as that of the Δ*hapR* strain (Fig. S4c). Thus, the Qrr1-4 sRNAs repress fractal morphogenesis via a mechanism that is independent of HapR. Consistent with this notion, *rbmA* expression in the *luxO*[D61E] Δ*hapR* Δ*qrr*1-4 mutant was four-fold higher than that in the *luxO*[D61E] Δ*hapR* strain, and indeed, was on par with that in the Δ*hapR* single mutant (Fig. 5g). Further verification for this Qrr1-4 role stems from the similar wrinkling fractal dimension values observed for the Δ*hapR* single mutant and the *luxO*[D61E] Δ*hapR* Δ*qrr*1-4 mutant (Fig. 5h). Likewise, the distributions of fine-scale structures were equivalent in the Δ*hapR* and the *luxO*[D61E] Δ*hapR* Δ*qrr*1-4 mutants (Fig. 5i). The pellicle morphologies of the QS mutants are summarized in Fig. 5j. We conclude that, although Qrr1-4 activate the first two stages of pellicle morphogenesis via repression of HapR production, they subsequently function in an opposing manner to suppress fractal wrinkling via a mechanism that is independent of HapR.

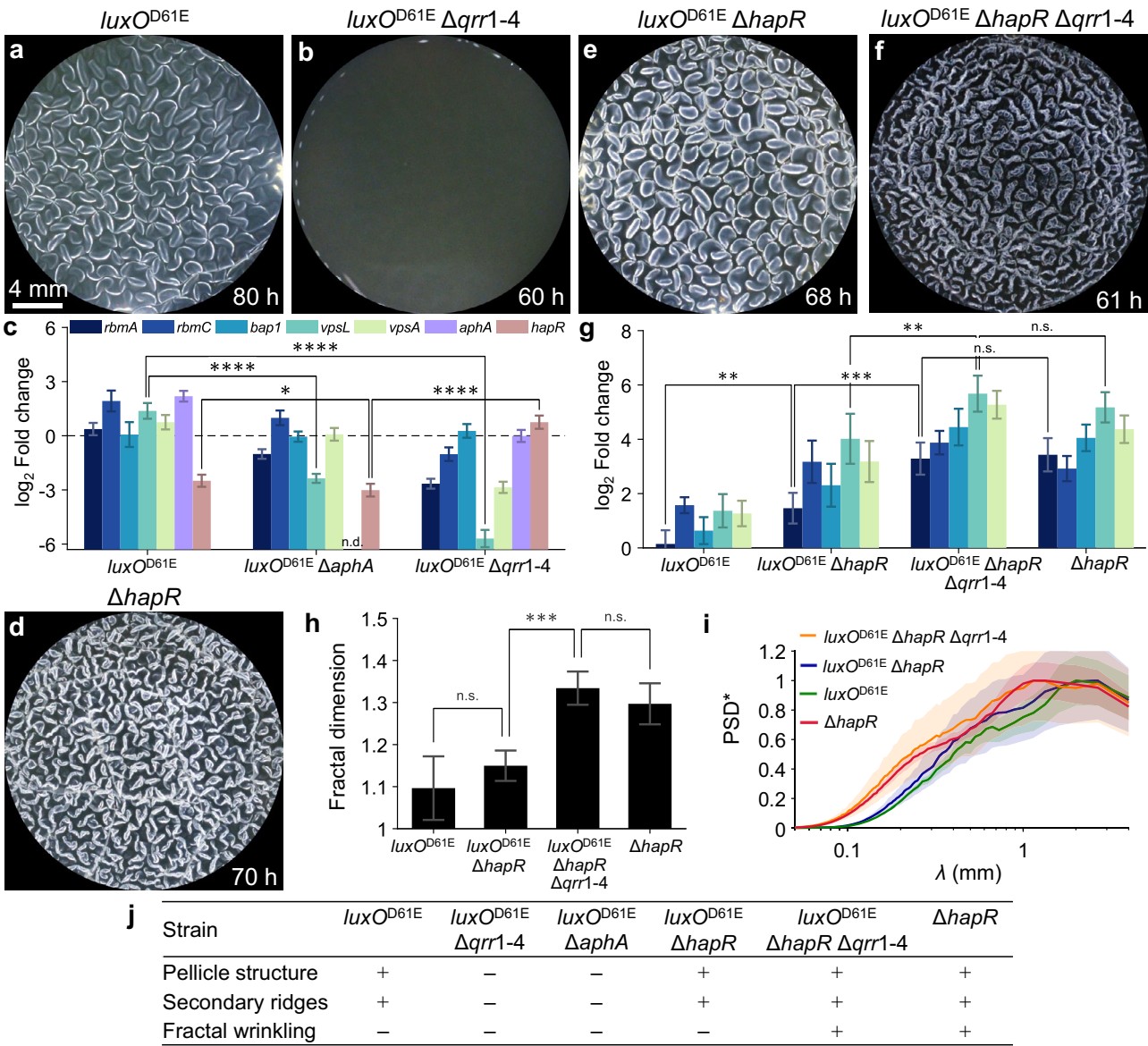

**Fig. 5 | *V. cholerae* QS mutants are defective in pellicle morphogenesis development and the Qrr sRNAs suppress fractal wrinkling by repressing genes encoding matrix proteins. a, b, d, e, f** Pellicle morphologies for the indicated strains at the indicated time points. **c** Expression of the indicated genes in the designated strains assessed by qRT-PCR. Data are presented relative to that from wildtype *V. cholerae*. **g** Expression profiles of key matrix genes in the designated QS mutants. Color scheme is shown in **c**. Data are presented relative to that from wildtype *V. cholerae*. **h** Fractal dimensions of wrinkle features for the mutants shown in **g** at the time points corresponding to the pellicles in **a**, **e**, **f**, **d**. **i** Distributions of wrinkling feature sizes for the mutants and conditions in **h**. Shaded areas denote standard errors from biological replicates (*n* = 3). **j** A summary of strain genotypes and their corresponding pellicle morphological phenotypes. In **c**, **g**, **h**, error bars denote standard deviations from biological replicates (*n* = 3). *P* values are: *, <0.05; **, <0.01; ***, <0.001; ****, <0.0001; n.s. (not significant), >0.05; and n.d., not detected.

## Overexpression of *qrr*2 eliminates fractal wrinkling via repression of *rbmA*

We next investigated the mechanism by which Qrr1-4 suppress fractal wrinkling. Given that RbmA acts downstream of the Qrr sRNAs (Fig. 1), we expected overexpression of *rbmA* in the low-cell-density locked *luxO*^D61E strain to override Qrr-mediated fractal wrinkling repression. Indeed, using the *luxO*^D61E mutant carrying the inducible p*rbmA* construct, we observed the emergence of fractal wrinkling following induction of *rbmA* with 0.002% arabinose (Fig. 6a, b). The fractal dimension $\delta = 1.3$ (Fig. 6d) achieved is comparable to that in the *luxO*^D61E Δ*hapR* Δ*qrr*1-4 strain (Fig. 5h). Therefore, with respect to modulation of fractal wrinkling, RbmA is epistatic to QS regulation. Next, we reintroduced Qrr2 via an inducible p*qrr*2 plasmid construct in the *luxO*^D61E Δ*hapR* Δ*qrr*1-4 mutant to test if fractal wrinkling could be synthetically repressed. Note

that Qrr1-4 function redundantly, and any one of them is sufficient to regulate target gene expression[38]. Here, Qrr2 was chosen for convenience. Fractal wrinkling decreased as inducer level increased (Fig. 6e–g), and fractal wrinkles were completely absent at an inducer concentration of 0.01% (Fig. 6g, Movie S9). Using qRT-PCR, we confirmed that overexpression of *qrr*2 in the *luxO*^D61E Δ*hapR* Δ*qrr*1-4 strain repressed expression of matrix genes, especially *rbmA* (Fig. 6h). Hence, we confirm that QS control of fractal wrinkling stems from Qrr1-4 post-transcriptional regulation of genes encoding matrix components including RbmA.

## Qrr sRNA-mediated repression of genes encoding matrix proteins requires VpsR but not VpsT

In *V. cholerae*, two transcription factors, VpsR and VpsT, activate matrix gene expression (Fig. 1). We tested whether Qrr1-4 regulate

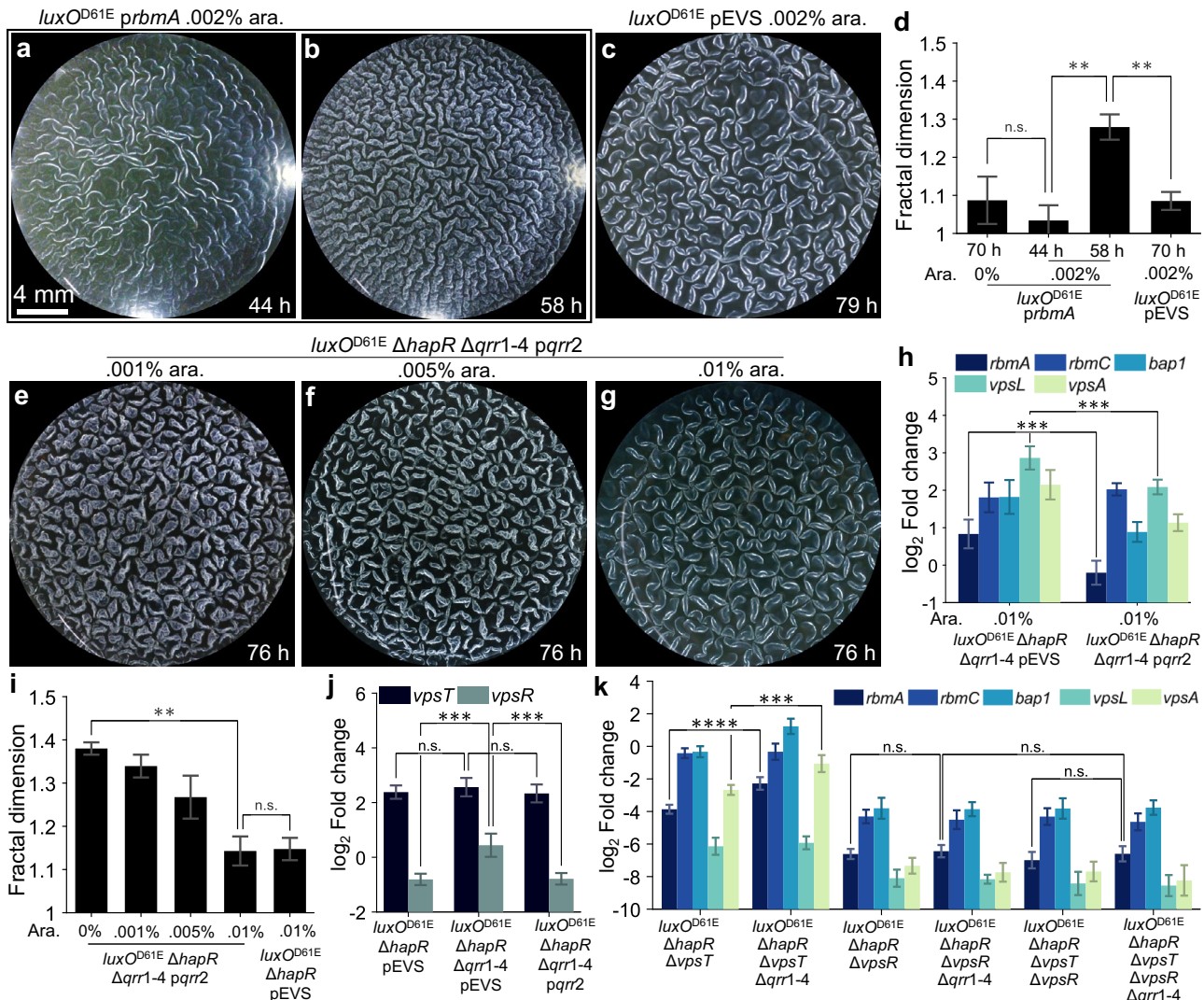

**Fig. 6 | Production of RbmA restores and Qrr2 represses fractal wrinkling in *V. cholerae* QS mutants. a,b** Pellicle morphologies for the *luxO*^D61E^ strain carrying p*rbmA* following induction with 0.002% arabinose were imaged at **a** 44 h in the ridge instability stage and **b** 58 h in the fractal wrinkling stage. **c** Pellicle morphologies for the *luxO*^D61E^ strain carrying the empty pEVS vector at 79 h. **d** Fractal dimensions of the wrinkle features for the *luxO*^D61E^ strain carrying p*rbmA* or the empty pEVS vector at the indicated time points and arabinose concentrations. **e–g** Pellicle morphologies for the *luxO*^D61E^ Δ*hapR* Δ*qrr*1-4 strain carrying p*qrr*2 following induction with the indicated arabinose concentrations at 76 h of growth. **h** Expression of the indicated genes in the designated strains was assessed by qRT-PCR. Data are presented relative to that from wildtype *V. cholerae*. **i** Fractal

dimensions of the wrinkle features for the *luxO*^D61E^ Δ*hapR* Δ*qrr*1-4 strain carrying p*qrr*2 or the empty pEVS vector following induction with the indicated arabinose concentrations. Data were taken following 76 h of growth. **j** Expression of the indicated genes in the designated strains assessed by qRT-PCR. Arabinose was provided at 0.01% in all cases. Data are presented relative to that from wildtype *V. cholerae*. **k** Expression of the indicated genes in the designated strains assessed by qRT-PCR. Data are presented relative to that from the *luxO*^D61E^ Δ*hapR* strain. In **d,h–k**, error bars denote standard deviations from biological replicates (*n* = 3). *P* values are: *, <0.05; **, <0.01; ***, <0.001; ****, <0.0001; n.s. (not significant), > 0.05; and n.d., not detected.

fractal wrinkling via repression of *vpsR* and/or *vpsT*. Compared to that in the *luxO*^D61E^ Δ*hapR* strain harboring the control vector, transcription of *vpsR* increased significantly in the *luxO*^D61E^ Δ*hapR* Δ*qrr*1-4 strain under pellicle forming conditions, and complementation with *qrr*2 was sufficient to repress *vpsR* (Fig. 6j). Deletion of *qrr*1-4 did not alter *vpsT* transcription. These results hint that the Qrr sRNAs function through VpsR but not VpsT to control pellicle morphogenesis. Indeed, transcription of the *rbmA* and *vpsA* genes were higher in the *luxO*^D61E^ Δ*hapR* Δ*vpsT* Δ*qrr*1-4 strain than in the *luxO*^D61E^ Δ*hapR* Δ*vpsT* strain (Fig. 6k), suggesting that the Qrr sRNAs continue to negatively regulate matrix synthesis in the absence of VpsT. However, this was not the case for VpsR, as transcription of all tested matrix genes were equivalent in the *luxO*^D61E^ Δ*hapR* Δ*vpsR* and the *luxO*^D61E^ Δ*hapR* Δ*vpsR* Δ*qrr*1-4 mutants (Fig. 6k). In fact, the absence of VpsR, irrespective of whether VpsT was

present or not, completely abolished Qrr1-4 regulation of matrix genes (Fig. 6k). Deletion of *qrr*1-4 in equivalent strains in the Rg background yielded similar but more subdued effects (Fig. S6). Therefore, Qrr sRNA-mediated repression of genes encoding matrix proteins requires VpsR, but not VpsT. Bioinformatic analyses suggest that Qrr1-4 have high sequence complementarity to a region spanning 180 to 230 bp downstream of the *vpsR* start codon, and there is weaker complementarity near the *vpsT* promoter (Fig. S7). However, preliminary experiments assaying Qrr2 control of a *vpsR-gfp* fusion in *Escherichia coli* did not show any effect on *vpsR* expression, suggesting Qrr regulation is indirect (Fig. S8). Nonetheless, it is clear that Qrr activity reduces matrix biosynthesis, and in so doing, suppresses the pellicle transition from the secondary ridge formation phase to the fractal wrinkling phase.

## Discussion

In this study, we report the molecular and regulatory underpinnings driving fractal wrinkling morphogenesis in *V. cholerae* pellicles at a fluid–fluid interface. Bacterial pellicles represent a model soft biomaterial that display key features such as active metabolism, growth, and self-patterning. In nature, these resilient bacterial communities may act as aquatic reservoirs for pathogenic bacteria, and pellicles are notorious in persistent infections and during disease transmission. Here, we reveal that at the molecular level, a pair of matrix proteins RbmC/Bap1, dictate whether the pellicle community stays afloat at the interface or self-peels from the meniscus. Second, only a single matrix protein, RbmA, is required to launch the *V. cholerae* fractal wrinkling morphogenesis program. Without RbmA, fractal wrinkling is abolished. Furthermore, artificial production of RbmA allows us to precisely tune the fractal dimensions of the pellicle and, moreover, to remodel mature pellicles already containing large-scale wrinkles and ridges. In this case, we can engineer mixed morphologies in which large features are decorated with fractal wrinkles. Such wrinkling structures with fractal scaling in wavelength, may offer possibilities for design of materials with enhanced surface areas to promote the diffusion and/or absorption of nutrients or other molecules of choice compared to materials harboring smooth wrinkles and ridges.

We revealed that production of the *V. cholerae* fractal pattern is orchestrated by QS via the regulatory Qrr1-4 sRNAs, which drive morphogenesis using two opposing mechanisms. First, the Qrr sRNAs launch the initial two wrinkling stages via repression of *hapR*, leading to activation of genes encoding matrix components. Second, the Qrr sRNAs suppress fractal wrinkling via a HapR-independent, VpsR-dependent mechanism. Hence, QS regulation and mutations that perturb the QS regulatory network propel *V. cholerae* into distinct morphologies. Thus, synthetic manipulation of QS could perhaps be used to promote or prevent *V. cholerae* pellicle maturation at fluid interfaces.

We note that in our current setup, total nutrient is fixed in the chamber at the start of the experiment and only limited nutrient remains after the pellicle achieves its mature morphology. Therefore, we do not expect strong differences in final biomasses among the mutants. An immediate future goal is to explore dynamic environments and the influence of fractal wrinkling on community development after maturation.

The core elements of bacterial pellicle morphogenesis, such as cell growth, matrix production, the accumulation of mechanical stresses, and morphogenic transformations are ubiquitous in both prokaryotic and eukaryotic multicellular systems. For example, in eukaryotes, folding of the gut, wrinkling of skin, and fractal branching of capillary blood vessels share many of the patterning and mechanical features we observe here in bacterial pellicles. Therefore, bacterial pellicles could provide tractable model systems to understand overarching principles underlying morphogenesis. Additionally, by controlling matrix constituents, nutrient acquisition, signaling gradients, or biomechanics, bacterial pellicles offer opportunities to engineer patterns for soft functional biomaterials.

## Methods

### Strains and growth media

*V. cholerae* strains used in this study are derivatives of the wildtype *V. cholerae* O1 biovar El Tor strain C6706. The rugose variant contains a missense mutation in the *vpvC* gene ($vpvC^{W240R}$, denoted Rg) that elevates c-di-GMP levels[23,39]. This strain forms robust biofilms on solid agar plates. QS mutants were constructed using the pKAS32 suicide vector[40]. Alterations were engineered into either the chromosome of wildtype *V. cholerae* or the Rg mutant, and whether wildtype or Rg *V. cholerae* is used as the parent or the strain for comparison is specified throughout. The pEVS143 plasmid was used for cloning genes under the arabinose-inducible *araC* promoter derived from the pBAD

vector[41]. A list of all strains used in this work is provided in Table S1. Lysogeny broth (LB) was used in all experiments and was supplemented with 50 μg/mL kanamycin for plasmid retention.

### Pellicle development

The *V. cholerae* pellicle growth protocol was adapted from that previously reported[19]. In brief, strains were grown overnight at 37 °C on LB agar plates with kanamycin when required for plasmid maintenance. A single colony was selected and resuspended in LB broth for 2 h with shaking at 37 °C. Kanamycin was included if plasmids were present. Cultures were grown to early exponential phase ($OD_{600} = 0.1–0.2$) and the cells were dispersed by vortex with 4 mm glass beads (Sigma) for 8 min. The cell density was measured, and the cultures were diluted to $OD_{600} = 0.01$. A total of 5.5 mL of culture inoculum was added to wells of 12-well plates so that the depth of liquid in each well was 14 mm. The cultures in the wells were overlaid with sterile mineral oil (Sigma Aldrich, volume 850 μL) to prevent evaporation and pellicle desiccation. To induce gene expression, arabinose was added at the time of inoculation or administered under the pellicle layer via a syringe needle (27 gauge, BD) as specified. Pellicles were allowed to develop at room temperature (20–21 °C) and were imaged from 15 h to 140 h as specified using a single lens reflex camera (Nikon D5100) mounted vertically and a 105 mm telephoto lens (Sigma, F2.8).

### Solid agar surface colony growth

Strains were grown overnight at 37 °C on LB agar plates. A single colony was selected and resuspended in LB broth for 2 h with shaking at 37 °C to early exponential phase ($OD_{600} = 0.1–0.2$). The cells were dispersed by vortex with 4 mm glass beads (Sigma) for 8 min. Aliquots of 1 μL were spotted onto LB agar plates and allowed to grow to the desired time points at 20-21 °C. Images of surface colony morphologies were taken with a Leica M205FA stereo microscope using a 1× plan apochromatic objective with numerical aperture 0.35.

### Imaging of pellicle interface detachment

Following the growth procedure above, 14 mL of culture diluted to a cell density of $OD_{600} = 0.01$ was placed in a sterile flask (Cellstar) with rectangular bottom cross-section (24 mm by 42 mm), so that the depth of liquid in the flask was 14 mm. The culture was covered with 2 mL of sterile mineral oil and allowed to grow at room temperature. The camera and lens setup was mounted sideways and focused on an imaging plane at the center of the flask.

### Image processing

Top-view time course images of pellicle morphologies were first registered in the $x–y$ plane using intensity-normalized cross-correlation methods. To extract wrinkle features, Frangi vesselness filtering[42,43] was applied to each image with a filter length scale corresponding to 1 pixel or 17 μm. The output vesselness intensities were used to generate power spectra and analyses of fractal dimensions.

### Fractal dimensions of pellicle surfaces

To quantify the fractal behavior and fractal dimensions of pellicle wrinkles, Frangi vesselness maps of pellicles were first binarized using intensity thresholding and then skeletonized using medial axis thinning algorithms[44]. Short and isolated branches in the skeletons were pruned. A standard box-counting algorithm was applied to the skeleton images. The scaling exponents of the box counts to the box sizes were obtained to define the fractal dimensions[45,46].

### RNA isolation and quantitative RT-qPCR

Cultures were grown as described above in the pellicle development procedure starting at $OD_{600} = 0.01$. After 24 h of growth at 20.5 °C, a volume of culture with cell number equivalent to 200 μL at $OD_{600}$ of 1 was collected and subjected to vortex with 4 mm beads for 8 min.

Samples were immediately exposed to two volumes of RNA protect reagent (Qiagen) and used for subsequent RNA isolation. After 15 min at room temperature, the cells were pelleted by centrifugation at 14,000 × g for 5 min and frozen at −80 °C. RNA was isolated using the RNeasy kit (Qiagen) and depleted of contaminating DNA using the TurboDNase (Applied Biosystems) kit. A total of 400 ng of DNase treated RNA was used to construct cDNA libraries using SuperScript IV Reverse Transcriptase (Invitrogen). RT-qPCR was performed using the PerfeCTa SYBR Green FastMix, Low ROX (Quanta Biosciences) reagent, and QuantStudio 6 Flex (Applied Biosystems). The manufacturers' recommended protocols were used. mRNA levels were normalized to that of the housekeeping topoisomerase gene *gyrA*. Primers used for RT-qPCR are provided in Table S2.

**Bioinformatic analyses**

A custom MATLAB (MathWorks) search algorithm was used to predict binding of Qrr sRNAs to *upsR* and *upsT* mRNAs. Briefly, the mRNA targets were scanned for regions complementary to regions of the Qrr sRNAs with a window size of 100 nucleotides. To identify regions of highest complementarity between sequences, local sequence alignments were performed using the Smith-Waterman (SW) algorithm[47]. The penalty for a non-complementary nucleotide was set to be 1/4 of the base-pairing score to capture sRNA-mRNA interactions with short regions of imperfect complementarity. To assess the significance of the predicted sRNA-mRNA base pairing, a null distribution of SW scores was generated by aligning the Qrr sRNA sequences to random mRNA sequences of the same length as the candidate mRNA targets. P-values were calculated by comparing the SW score of the actual alignment to the null distribution and adjusting for multiple testing.

**Reporting summary**

Further information on research design is available in the Nature Research Reporting Summary linked to this article.

## Data availability

Source data are provided as a Source Data file. Source data are provided with this paper.

## Code availability

Source codes are available online at Zenodo, https://doi.org/10.5281/zenodo.7130657.

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

## Acknowledgements

We thank Ameya M. Mashruwala for help with qRT-PCR. We thank Chenyi Fei for performing the complementarity analysis. We thank members of the Bassler group, Howard A. Stone, and Ned S. Wingreen for insightful discussions. We thank Ned S. Wingreen, Ameya M. Mashruwala, and Chenyi Fei for carefully reading the manuscript. This work was supported by the Howard Hughes Medical Institute, National Science Foundation grants MCB-2043238 and 1853602, and NIH grant 5R37GM065859 to B.L.B.

## Author contributions

B.Q. and B.L.B. designed experiments; B.Q. performed experiments; B.Q. and B.L.B analyzed data and wrote the manuscript.

## Competing interests

The authors declare no competing interests.
