## [Peer Review File · Nature Communications]

Quorum-sensing control of matrix protein production drives fractal wrinkling and interfacial localization of *Vibrio cholerae* pelliclesReviewer #1 (Remarks to the Author):

The authors report the molecular and regulatory underpinnings driving fractal wrinkling morphogenesis in *V. cholerae* pellicles at a fluid-fluid interface. They reveal that at the molecular level, a pair of matrix proteins RbmC/Bap1, dictate whether the pellicle community stays afloat at the interface or self-peels from the meniscus. They find that only a single matrix protein, RbmA, is required to launch the *V. cholerae* fractal wrinkling morphogenesis program. They reveal that production of the *V. cholerae* fractal pattern is orchestrated by QS via the regulatory Qrr 1-4 sRNAs, which drive morphogenesis using two opposing mechanisms.

My comments on this article are as follows:

1. How is "quorum sensing" defined in the article? How do RbmA, RbmC and Bap1 relate to the quorum sensing?
2. Are "low-cell-density" and "high-cell-density" two modes of quorum sensing? What is the difference between these two modes?
3. What is the relationship between Qrr1-4 sRNAs and quorum sensing
4. How do the authors demonstrate the effect of quorum sensing on fractal wrinkling? Which figures represent the effects of quorum sensing?
5. The "quorum sensing control" mentioned in the title of the article has no corresponding representation in the text.
6. What does the ordinate "Fold change" in Fig. 3k mean? Is there an explanation in the text?
7. There are too many subtitles and too much content in the Results section of the article, which makes the structure of the article confusing.
8. Experiment is the main content of this paper, but the experimental phenomenon is superficial and the research depth is not enough.
9. This article is too long and may not fit the style of Nature Communications.

Reviewer #2 (Remarks to the Author):

This is a very precise and clearly written paper from the Bassler lab. While reading the first half of the results, it appeared slightly descriptive, in addition to be highly quantitative on how different *Vibrio* matrix components are contributing to wrinkle formation of pellicles and colonies. While this is new, but lacks true novelty, as matrix components have been described in other organisms to contribute to pellicle wrinkle formation, and the different nature of *Vibrio* matrix proteins has been also extensively studied previously. Nevertheless, the last three chapters of the results brings novelty to this work, describing new molecular microbiology details behind connection between QS, i.e. Qrr sRNAs and transcriptional regulation of matrix expression.

In general, the authors should be careful and explain more the fractal wringing and how the detected parameters describing the degree of wrinkle formation, as many readers will come with a microbiology background and therefore the biophysical aspects might be more complicated to understand.

One experimental detail missing here is the interaction between Qrr sRNA and *vpsR* RNA, which could be tested by mutating specific bases of the dedicated region within *vpsR* without influencing the coded amino acid sequence. This would provide a strong proof on the proposed mechanism behind this nicely described cascade. Alternatively, *in vitro* interaction would be needed as a proof on the proposed mechanisms that is currently solely based on bioinformatics.

Please explain in more details whether over expression of any of the Qrr will have identical phenotypes and therefore why Qrr2 was selected. There is a reference indicated, but possibly more details are needed for explanation.

Minor comments:

Abstract, line 12: "fractal scaling in wavelength" the term should be explained for readers with microbiology background

p3, line 40: remove one of the word from "occurs correlates "

p4, line 21: the cited works are all from the Bassler lab (in addition to one example that describes

vpvC mutation, but not placing most experiment on that mutant only); please cite more (independent, thus not Bassler lab and offsprings from the same lab) papers that mainly use vpvC mutant. Alternatively, just state "As we have done in the past"

p5, line 1: sentence is highly speculative, rather use: "This feature might contribute to nutrient capture and cell growth enhancement."

p6, line 38: correct to "survival advantages at the population level." , this study does not examine communities (=more than one species)

p26, line 28: correct "Soldi"

Reviewer #3 (Remarks to the Author):

Qin and Bassler identify regulatory and structural components of pellicle morphogenesis in *Vibrio cholerae*. Through a thorough series of quantitative imaging experiments with mutants, conditions, and strain backgrounds, the authors isolate particular structural molecules and new regulatory relationships which lead to a fractal wrinkling pattern, even going so far as to tune the fractal dimension and control its temporal developmental course. Overall, we found the data impressive and the paper a valuable contribution with broad interest. Figure 4 was an especially strong result. We did find that the paper became a struggle to follow once we hit the quorum sensing section. Below, we will outline some questions and suggestions that we hope can make the key messages of the paper clearer. To reiterate, we find the results impressive and the conclusions justified. We would like to help make the message easier to follow. We did our best with the three major points below, followed by several minor suggestions.

- Generally the section of the paper associated with Fig. 5 was difficult to follow. During review, we discussed ways to streamline it, without coming up with any tremendous ideas, so we understand the challenge. This suggestion may be crude, but we found a genotype/phenotype table useful. There are strains containing combinations of the LuxOD16E, qrr1-4, and hapR. Those strains exhibit interface localization, ridge formation, and/or fractal wrinkling. Laying out those genotypes and phenotypes in a table made it easier to reason through the regulatory relationships. The text itself we could follow, but whenever we looked at the data in Fig. 5, we got lost.
- For a researcher outside the field, Fig. 1 is daunting. In such a broad journal, it would be great to have some straightforward, key information in the actual figure. Do you have images that can quickly demonstrate what interface localization and wrinkling are, as well as the fact that you are investigating bacterial biofilms? Furthermore, would it be possible to add annotations to the important aspects of the diagram itself, for example that LuxO integrates QS information. Can you show that RbmC, Bap1, RbmA, and VPS are structural products and not transcription factors or other gene expression regulators? Despite its relevance to the Rg strain, we also thought that the c-di-GMP arrow was perhaps extraneous information that obscured the key information.
- What is the connection between fractal dimension and the PSD spectra? For a given spectrum, would you expect a particular fractal dimension? We often didn't know what to take from the PSD graphs. For example, in Fig. 3i, the 0.005% condition has a similar fractal dimension to the 0.001% condition, but the two spectra in Fig. 3j are quite different. Is that expected? What should we make of that? Some elaboration on the PSD graphs and how they relate to fractal wrinkling would be useful.

Smaller points:

- Despite the experimental system being described in a previous paper, we suggest including 1-2 sentences describing the specific experimental setup and the liquid-liquid interface, perhaps near page 3, line 26.
- What are the precise criteria for a wrinkled interface to be considered "fractal"? There is a brief mention of this on page 4, line 44, but it says " $d \sim 1$ " means something is not a fractal. Did you have a quantitative criterion?
- If possible, we would suggest removing the left part of 2k from the figure, but keeping the movie. We found the righthand time series of 2k excellent, but had trouble making sense of the left image.
- What is ϵ in Fig. 2g? The panel has $1/\epsilon$, but the legend has ζ .
- With some images in the paper, multiple time points of the same colony are included with

individual snapshots of varying strains. The reader must put in extra effort to understand what is shown in the figure. Is it possible to subtly identify time series images, perhaps by giving them the same panel letter within the figure and/or outlining them? Examples are Fig. 4a-c, Fig. 4d-f, and Fig. 6a-b.

- Multiple regulators are referred to as "master" regulators. This may contribute to the challenge in following the logic of the quorum sensing section. Can only one (or none) regulator be referred to as a "master" regulator, even if they are regulators of different things?
- The section entitled "QS regulates progression through the *V. cholerae* pellicle fractal wrinkling program" discusses a strain that is later revealed to not exhibit fractal wrinkling. This was confusing. Our suggestion would be to open this section summarizing the necessary steps of fractal wrinkling and mention that they are/may be regulated by QS. This can also serve as a transition into section of the paper on QS regulation, which is challenging to follow as we mentioned above.
- Can you add a citation for Rg having high levels of c-di-GMP?
- The vast majority of strains in the paper are locked into the low-density QS state or isolated from QS regulation. Does the high-density state come into play late in development?

Reviewer #4 (Remarks to the Author):

Qin and Bassler investigated the structural components driving pattern formation in the pellicles of *Vibrio cholerae*. They find that matrix proteins and a quorum-sensing regulator drive the formation of the fractal patterns in pellicles. In particular, the proteins RbmC and Bap1 allow the formation of stable pellicles at the fluid interface, while the protein RbmA controls the formation of the fractal patterning. Furthermore, the expression level of RbmA controls the wavelength of the wrinkles and, when expressed at a level higher than the one normally present in the used strain, suppresses wrinkling. Lastly, the role of quorum sensing is elucidated, and two opposite mechanisms are identified to trigger and stop the patterns' formation.

The paper is well written, and the quality of the images is appropriate for publication in Nature Communications. Furthermore, the data strongly supports the conclusions of this paper on the structural components and the molecular mechanisms controlling pattern formation. Still, the impact of the pellicles composition on their material properties and, consequently, on the wrinkle formation needs to be elucidated before publication.

1. In the PNAS, 2021, the same authors develop a mathematical model to relate the pattern's emergent wavelength to its bending stiffness of the film. In the current work, what is the impact of the RbmC, Bap1, and RbmA proteins on the film mechanics? In the case of the RbmA protein, the differences in mechanical properties induced by different expression levels should also be probed. Given the amount of data already presented by the authors, I am not asking for an in-depth investigation. However, some trends in the relation between film composition/mechanics/fractal wrinkles should be investigated for completeness.
2. How does fractal wrinkling affect biomass accumulation and pellicles thickness? Could the thickness of the pellicles be quantified and compared between the different compositions studied?
3. A minor comment regarding the style: while molecular and gene expression mechanisms are described in detail, a consistent part of the quantification of the physical parameter describing the fractal pattern is confined in the figure caption. I would suggest adding some more details about these results in the text.

Reviewer #1 (Remarks to the Author):

The authors report the molecular and regulatory underpinnings driving fractal wrinkling morphogenesis in *V. cholerae* pellicles at a fluid-fluid interface. They reveal that at the molecular level, a pair of matrix proteins RbmC/Bap1, dictate whether the pellicle community stays afloat at the interface or self-peels from the meniscus. They find that only a single matrix protein, RbmA, is required to launch the *V. cholerae* fractal wrinkling morphogenesis program. They reveal that production of the *V. cholerae* fractal pattern is orchestrated by QS via the regulatory Qrr 1-4 sRNAs, which drive morphogenesis using two opposing mechanisms.

My comments on this article are as follows:

1. How is "quorum sensing" defined in the article? How do RbmA, RbmC and Bap1 relate to the quorum sensing?

Quorum sensing (QS) refers to the process of cell-to-cell communication that bacteria use to orchestrate collective behaviors. QS relies on the production, release, and group-wide detection of signal molecules, called autoinducers. As shown in Fig.1, signaling through the QS pathway alters the phosphorylation state of the key QS regulator protein LuxO, which activates expression of genes encoding four regulatory small RNAs to modulate expression of a host of genes. One crucial set of QS-controlled target genes encode the matrix components, including RbmA, RbmC, Bap1, and VPS.

2. Are "low-cell-density" and "high-cell-density" two modes of quorum sensing? What is the difference between these two modes?

Yes. "low-cell-density" refers to QS when cells are alone and act as individuals, and "high-cell-density" refers to the opposite case, when bacteria are in communities and act as members of collectives.

3. What is the relationship between Qrr1-4 sRNAs and quorum sensing

As described in the text and in Fig. 1, at the core of the *Vibrio cholerae* QS signal transduction pathway lie four small RNAs (sRNAs), named the Quorum Regulatory RNAs (Qrr1–4), that relay the QS signal to the downstream target genes.

4. How do the authors demonstrate the effect of quorum sensing on fractal wrinkling? Which figures represent the effects of quorum sensing?

We show in this work that QS signaling components modulate pellicle formation and fractal wrinkling via the following mechanism: First, phosphorylated LuxO at low-cell-density activates the expression of the genes encoding the sRNAs Qrr1-4. The Qrr sRNAs drive AphA production and AphA turns on biofilm synthesis (Fig. 5b, Fig. S5).

Second, the Qrr sRNAs repress production of the high-cell-density master regulator HapR. That function prevents matrix degradation and inhibits dispersal of the cells from the community. Thus, QS, acting through the Qrr sRNAs, is required for robust pellicle formation (Fig. 5a,e) but not sufficient for fractal wrinkling morphology. What is surprising, as we have uncovered here (Fig. 5, 6), is that the Qrr1-4 sRNAs impede fractal wrinkling in a HapR independent manner. We show that occurs via repression of the VpsR biofilm master regulator (Fig. 6j,k). That is why elimination of the Qrr1-4 sRNAs restores fractal wrinkling in the *luxOD47E ΔhapR* mutant (Fig. 5f). Thus, QS drives the global morphology of the community and its fractal pattern.

5. The "quorum sensing control" mentioned in the title of the article has no corresponding representation in the text.

We note that QS control (of the wrinkling pattern) is the key finding in this work and the results are described in the following subsections:

"QS regulates progression through the *V. cholerae* pellicle fractal wrinkling program."

"The QS Qrr sRNAs suppress *V. cholerae* pellicle fractal wrinkling via a HapR-independent mechanism."

"Overexpression of *qrr2* eliminates fractal wrinkling via repression of *rbmA*."

"Qrr sRNA-mediated repression of genes encoding matrix proteins requires VpsR but not VpsT."

The relevant figures that demonstrate QS control are: Figs. 5, 6, S4, S5, S6, S7.

6. What does the ordinate "Fold change" in Fig. 3k mean? Is there an explanation in the text?

Fold changes are the standard units one uses to represent qRT-PCR results in the literature. For reference, to obtain the fold change, we compute the arithmetic difference between qRT-PCR threshold amplification cycle counts for test genes compared to that of the control house-keeping gene *gyrA*. When gene expression is compared between strains, results are shown on a base 2 logarithmic scale, hence, the \log_2 Fold change nomenclature is used. In response to this point, we have added the following clarification to the qRT-PCR methods section, page 27, line 19:

"mRNA expression levels were normalized to the housekeeping topoisomerase gene *gyrA*".

7. There are too many subtitles and too much content in the Results section of the article, which makes the structure of the article confusing.

With respect, we disagree. The subsection titles follow standard journal style and they help to orient readers to the central topics.

8. Experiment is the main content of this paper, but the experimental phenomenon is superficial and the research depth is not enough.

With respect, we disagree. The results reported here are novel and represent important discoveries for the field. Moreover, they span from highly quantitative analyses of community morphology, to the level of the biological signaling cascade, to the precise molecular components involved. For the first time, we establish the link between matrix elements and the cell-cell signaling components that control them to the macroscopic morphology of the bacterial community, a fundamental advance for the field.

9. This article is too long and may not fit the style of *Nature Communications*.

We respectfully disagree with the reviewer on these points. We have used a host of molecular biology techniques, including mutagenesis, qRT-PCR, time-resolved induction of target genes, bioinformatics, as well as time course imaging with various modalities, and pellicle morphology image analyses to conclusively pinpoint the molecular and QS signaling mechanisms driving bacterial community patterning. Our results connect cell-cell signaling and architectural components to morphogenic patterning and suggest that manipulation of QS regulators or synthetic control of *rbmA* expression could underpin strategies to engineer soft biomaterial morphologies on demand. Our results represent a significant advance in the understanding of multicellular morphogenesis. Thus, the contributions here are appropriate for publication in *Nature Communications* and are of interests to the general audience of biologists, engineers, and material physicists.

Reviewer #2 (Remarks to the Author):

This is a very precise and clearly written paper from the Bassler lab. While reading the first half of the results, it appeared slightly descriptive, in addition to be highly quantitative on how different *Vibrio* matrix components are contributing to wrinkle formation of pellicles and colonies. While this is new, but lacks true novelty, as matrix components have been described in other organisms to contribute to pellicle wrinkle formation, and the different nature of *Vibrio* matrix proteins has been also extensively studied previously. Nevertheless, the last three chapters of the results brings novelty to this work, describing new molecular microbiology details behind connection between QS, i.e. Qrr sRNAs and transcriptional regulation of matrix expression.

We thank the Reviewer for the positive comments concerning the rigor and novelty of our work.

In general, the authors should be careful and explain more the fractal wringing and how the detected parameters describing the degree of wrinkle formation, as many readers will come with a microbiology background and therefore the biophysical aspects might be more complicated to understand.

As suggested by the Reviewer, we have incorporated more explanatory text to clarify the link between physical parameters concerning wrinkling and the degree of wrinkling. For example, we have included the following text to explain the fractal dimension at page 4 line 44:

“A fractal scaling large than 1 indicates increased complexity in the surface geometry of the wrinkle features.”

And at page 6 line 1:

“...the fractal dimensions of the wrinkling features also increased (Fig. 3i), indicating the emergence of self-similar wrinkles and an increase in their geometric complexity. Additionally, the power spectra density (PSD) of the wrinkling features shifted toward smaller wavelengths (Fig. 3j), confirming the emergence of small-sized wrinkling features.”

One experimental detail missing here is the interaction between Qrr sRNA and *vpsR* RNA, which could be tested by mutating specific bases of the dedicated region within *vpsR* without influencing the coded amino acid sequence. This would provide a strong proof on the proposed mechanism behind this nicely described cascade. Alternatively, *in vitro* interaction would be needed as a proof on the proposed mechanisms that is currently solely based on bioinformatics.

We thank the reviewer for this good idea! Indeed, in the previous version, the region of interaction was predicted but not proven experimentally. To further probe the interaction between the Qrr sRNA and *vpsR*, we conducted two additional experiments. First, we overexpressed a Qrr sRNA (Qrr2 as our representative sRNA) in the *luxO*^{D61E} Δ *hapR* Δ *qrr1-4* mutant and monitored the expression of the *vpsT* and *vpsR* targets via qRT-PCR (revised Fig. 6j). As anticipated from the results shown in Figs. 5 and 6, a significant decrease in *vpsR* expression occurred following induction of *qrr2*. This result shows that the Qrr sRNA is both necessary and sufficient to repress the target *vpsR* gene. Accordingly, page 9 line 26 has been edited to read:

“We tested whether Qrr1-4 regulate fractal wrinkling via repression of *vpsR* and/or *vpsT*. Compared to that in the *luxO*^{D61E} Δ *hapR* strain harboring the control vector, transcription of *vpsR* increased significantly in the *luxO*^{D61E} Δ *hapR* Δ *qrr1-4* strain under pellicle forming conditions, and complementation with *qrr2* was sufficient to repress *vpsR* (Fig. 6j). Deletion of *qrr1-4* did not alter *vpsT* transcription. These results hint that the Qrr sRNAs function through VpsR but not VpsT to control pellicle morphogenesis.”

Second, to test whether Qrr regulation of *vpsR* is direct, i.e., that the sRNA and the mRNA interact with one another, we constructed a vector carrying inducible *qrr2* and another vector carrying *vpsR* fused to *gfp*. The cloned region of *vpsR* included the putative Qrr binding region we had predicted and shown in Fig. S7. As the referee suggested, we also mutated the putative Qrr binding region by randomizing the nucleotides encoding the site in an analogous *gfp* fusion construct (designated *vpsR*^{*}). We transferred these constructs into *E. coli* to eliminate any possible participation by other *V. cholerae* components. We then measured GFP production following Qrr induction. We note that we have used this exact strategy many times to distinguish between direct and indirect Qrr sRNA regulation of target genes (see for example, Papenfort *et al*, *PNAS* 112.7 (2015): E766-E775). Our results suggest that induction of *qrr2* does not affect *vpsR-gfp* expression, as shown in new Fig. S8. This finding suggests an indirect regulatory mechanism. Companion text explaining the new results are on page 9 line 41:

“However, preliminary experiments assaying Qrr2 control of a *vpsR-gfp* fusion in *Escherichia coli* did not show any effect on *vpsR* expression, so that analysis did not yield evidence for a direct interaction between the Qrr and *vpsR* (Fig. S8). Nonetheless, it is clear that Qrr activity (Fig. S8). Nonetheless, it is clear that Qrr activity reduces matrix biosynthesis, and in so doing, suppresses the pellicle transition from the secondary ridge formation phase to the fractal wrinkling phase.”

Given that the mechanism of Qrr regulation of *vpsR* is apparently indirect and thus requires additional factors that are unknown, we hope the Reviewer will understand that characterizing the new pathway connecting the Qrr sRNAs to *vpsR* in its entirety is beyond the scope of this paper. Such an undertaking, while fascinating, will require a mutant screen to reveal component(s) that act between the Qrr sRNAs and the *vpsR* transcript, followed by characterization of their functions, binding interactions, and

regulatory mechanisms. Here, we have shown (Figures 5, 6, and S6) that the Qrr sRNAs are responsible for repressing *vpsR* and, in turn, pellicle fractal patterning. Our discovery and establishment of a role for the Qrr sRNAs in this morphogenic process is but one finding among several key discoveries. Thus, we ask that the full characterization of this new Qrr pathway and its components be left for a future stand-alone work.

Please explain in more details whether over expression of any of the Qrr will have identical phenotypes and therefore why Qrr2 was selected. There is a reference indicated, but possibly more details are needed for explanation.

The regulatory sRNAs are functionally redundant. They all control the same downstream target genes, and they can substitute for one another. Moreover, they are subject to dosage compensation, so if we remove three of the *qrr* genes, expression of the remaining *qrr* gene increases to compensate (Svenningsen *et al*, *EMBO J.* 28.4 (2009): 429-439). We used Qrr2 as the representative Qrr for convenience. Identical results would occur if we used another Qrr sRNA, which we showed in the previous Svenningsen *et al* study. To clarify, we have edited page 9 line 3 to read:

“Note that Qrr1-4 function redundantly and any one of them is sufficient to regulate target gene expression³⁸. Here, Qrr2 was chosen for convenience.”

Minor comments:

Abstract, line 12: "fractal scaling in wavelength" the term should be explained for readers with microbiology background.

To clarify this point, we have included the following explanatory text on fractal dimensions at page 4 line 44:

“A fractal scaling large than 1 indicates increased complexity in the surface geometry of the wrinkle features.”

p3, line 40: remove one of the word from "occurs correlates "

Good catch. The sentence (page 3 line 40) now reads:

“Indeed, inducible expression of *rbmA* in a Δ *rbmA* mutant triggers rapid formation of fractal wrinkles, and the level of fine-scale wrinkling correlates with the strength of *rbmA* expression.”

p4, line 21: the cited works are all from the Bassler lab (in addition to one example that describes *vpvC* mutation, but not placing most experiment on that mutant only); please cite more (independent, thus not Bassler lab and offsprings from the same lab) papers that mainly use *vpvC* mutant. Alternatively, just state "As we have done in the past"

Yes, thank you for catching our omission. Indeed, that strain is used by many labs and it did not originate with us. The sentence now reads:

“As we and others have done in the past^{14,17,19,23}, we rely on a commonly used hyper-matrix-producing *V. cholerae* strain carrying the *vpvC*^{W240R} mutation.”

p5, line 1: sentence is highly speculative, rather use: "This feature might contribute to nutrient capture and cell growth enhancement."

Yes, a good change. The sentence now reads:

“This feature may contribute to the capture of nutrients and enhanced cell growth²⁹.”

p6, line 38: correct to "survival advantages at the population level." , this study does not examine communities (=more than one species)

Indeed, a more appropriate term is needed. The sentence now reads:

“ ... could be energetically economical, providing survival advantages at the population level.”

p26, line 28: correct "Soldi"

We corrected the typo. We thank the Reviewer for the especially careful reading of the manuscript.

Reviewer #3 (Remarks to the Author):

Qin and Bassler identify regulatory and structural components of pellicle morphogenesis in *Vibrio cholerae*. Through a thorough series of quantitative imaging experiments with mutants, conditions, and strain backgrounds, the authors isolate particular structural molecules and new regulatory relationships which lead to a fractal wrinkling pattern, even going so far as to tune the fractal dimension and control its temporal developmental course. Overall, we found the data impressive and the paper a valuable contribution with broad interest. Figure 4 was an especially strong result. We did find that the paper became a struggle to follow once we hit the quorum sensing section. Below, we will outline some questions and suggestions that we hope can make the key messages of the paper clearer. To reiterate, we find the results impressive and the conclusions justified. We would like to help make the message easier to follow. We did our best with the three major points below, followed by several minor suggestions.

We thank the Reviewer for the positive comments about the novelty and quality of our work.

- Generally the section of the paper associated with Fig. 5 was difficult to follow. During review, we discussed ways to streamline it, without coming up with any tremendous ideas, so we understand the challenge. This suggestion may be crude, but we found a genotype/phenotype table useful. There are strains containing combinations of the LuxOD16E, qrr1-4, and hapR. Those strains exhibit interface localization, ridge formation, and/or fractal wrinkling. Laying out those genotypes and phenotypes in a table made it easier to reason through the regulatory relationships. The text itself we could follow, but whenever we looked at the data in Fig. 5, we got lost.

We thank the Reviewer for this idea for clarification. As suggested, we have added a genotype/phenotype table to Fig. 5 to guide the reader. The table also summarizes the key results/takeaways from the figure.

- For a researcher outside the field, Fig. 1 is daunting. In such a broad journal, it would be great to have some straightforward, key information in the actual figure. Do you have images that can quickly demonstrate what interface localization and wrinkling are, as well as the fact that you are investigating bacterial biofilms? Furthermore, would it be possible to add annotations to the important aspects of the diagram itself, for example that LuxO integrates QS information. Can you show that RbmC, Bap1, RbmA, and VPS are structural products and not transcription factors or other gene expression regulators? Despite its relevance to the Rg strain, we also thought that the c-di-GMP arrow was perhaps extraneous information that obscured the key information.

Yes, we agree that the QS circuit may be difficult to follow for non-specialists. Hence, as suggested, we have added text to the revised Fig. 1 to guide the reader about the roles

of the main components studied in this work. Regarding adding a picture or cartoon for interfacial localization, we do not think a cartoon is better than the real data/images that are already in Fig. 2. Lastly, per the Reviewer's suggestion, we have removed the c-di-GMP signaling input in the scheme as it is not the focus of the current investigation, and it makes the figure unnecessarily complicated.

• What is the connection between fractal dimension and the PSD spectra? For a given spectrum, would you expect a particular fractal dimension? We often didn't know what to take from the PSD graphs. For example, in Fig. 3i, the 0.005% condition has a similar fractal dimension to the 0.001% condition, but the two spectra in Fig. 3j are quite different. Is that expected? What should we make of that? Some elaboration on the PSD graphs and how they relate to fractal wrinkling would be useful.

To clarify, the fractal dimension and the PSD characterize different morphological features. The fractal dimension measures (1) the presence of self-similar wrinkling structures that span many length-scales and (2) the geometric complexity of such structures. A high fractal dimension implies more complex features (as defined by the scaling in dimensional box counts) than a low fractal dimension. The power spectra density (PSD) as a function of wrinkling wavelength, on the other hand, describes the overall size distribution of the wrinkling features. Specifically, the PSD allows us to discern whether there is bias in the distribution. In principle, it is possible to have no fractal dimension but a bias toward small sized features. In that case, there would be an absence of self-similar features spanning multiple length scales.

To clarify this important point, we have edited page 6 line 1 to read:

"...the fractal dimensions of the wrinkling features also increased (Fig. 3i), indicating the emergence of self-similar wrinkles and an increase in their geometric complexity. Additionally, the power spectra density (PSD) of the wrinkling features shifted toward smaller wavelengths (Fig. 3j), confirming the emergence of small-sized wrinkling features."

And to page 6 line 29, we added the following text:

"Together, these results show that overexpression of *rbmA* generates small size wrinkles and eliminates the self-similar cascade of wrinkles that span small to large wavelengths."

Smaller points:

• Despite the experimental system being described in a previous paper, we suggest including 1-2 sentences describing the specific experimental setup and the liquid-liquid interface, perhaps near page 3, line 26.

As suggested, we have added the following sentence to page 3, line 26:

“Using a custom stereomicroscope setup that adaptively tracked morphological features, we characterized the sequential morphological stages of *V. cholerae* pellicles as they formed at the interface between the growth medium and mineral oil.”

• What are the precise criteria for a wrinkled interface to be considered “fractal”? There is a brief mention of this on page 4, line 44, but it says “ $d \sim 1$ ” means something is not a fractal. Did you have a quantitative criterion?

We regret the confusion. To clarify, the fractal dimension is characterized on a topological skeleton of each connected wrinkle feature and then averaged over all such individual features. Hence, a simple curved wrinkle, such as a ridge, or straight wrinkling, such as the primary wrinkle, will yield a Hausdorff dimension of 1. This outcome occurs because those features lack branching and self-similar patterning in their topologies. Fractal wrinkles, on the other hand, will yield a much higher dimension due to the connected cascade of fine scale wrinkles. As the Reviewer pointed out, there is no absolute threshold for what is and is not fractal, but here, we can measure large and statistically significant differences between the various mutants.

• If possible, we would suggest removing the left part of 2k from the figure, but keeping the movie. We found the righthand time series of 2k excellent, but had trouble making sense of the left image.

To clarify, the space-time kymograph on the left side of Fig. 2k shows the rapid and often catastrophic self-peeling of the $\Delta rbmC$ and $\Delta bap1$ mutant pellicles that occurs over a short time scale. Therefore, these findings are integral to the figure and its message. To assist the reader in interpreting Fig. 2k left, as the Reviewer suggests, we added the following text to page 5 line 20:

“The pellicle subsequently self-peels from the interface and catastrophically collapses over a short time scale (Fig. 2k, i).”

• What is ϵ in Fig. 2g? The panel has $1/\epsilon$, but the legend has ζ .

We thank the Reviewer for pointing out this typo. The symbol ϵ represents box lengths.

• With some images in the paper, multiple time points of the same colony are included with individual snapshots of varying strains. The reader must put in extra effort to understand what is shown in the figure. Is it possible to subtly identify time series images, perhaps by giving them the same panel letter within the figure and/or outlining them? Examples are Fig. 4a-c, Fig. 4d-f, and Fig. 6a-b.

We thank the Reviewer for this good suggestion. We have incorporated these visual guides.

- Multiple regulators are referred to as “master” regulators. This may contribute to the challenge in following the logic of the quorum sensing section. Can only one (or none) regulator be referred to as a “master” regulator, even if they are regulators of different things?

Again, we regret the confusion. Indeed, there can be more than one master regulator if each regulator functions during a unique cell state. In the case of QS, AphA is the master regulator at low-cell-density and HapR is the master regulator at high-cell-density. To avoid confusion, we clarify their roles and we now call LuxO a central regulator because it works at the heart of the system under all cell density states.

- The section entitled “QS regulates progression through the *V. cholerae* pellicle fractal wrinkling program” discusses a strain that is later revealed to not exhibit fractal wrinkling. This was confusing. Our suggestion would be to open this section summarizing the necessary steps of fractal wrinkling and mention that they are/may be regulated by QS. This can also serve as a transition into section of the paper on QS regulation, which is challenging to follow as we mentioned above.

The low-cell-density QS state drives the key initial stage of fractal wrinkling, namely the formation of confluent cell layers and the onset of curved secondary ridges. That is why we used “progression” in this sub-title. We also chart the sequential activities of the biocomponents crucial to each of the four stages of morphogenesis, leading to fractal wrinkling. We use mutants that do not display fractal wrinkling to pinpoint which component(s) is required in each step. The comparisons of morphologies that we show are central to our argument about the roles of the different biocomponents. Moreover, showing the differences between mutant and wildtype phenotypes (in this case, morphology) is widely accepted as proof to underpin genetic analyses.

To assist the reader in following our logic, as the Reviewer suggested, we have added the following text to page 7 line 38:

“Next, we investigated the key QS components that control the morphological stages leading to fractal wrinkling.”

- Can you add a citation for Rg having high levels of c-di-GMP?

The cytoplasmic c-di-GMP measurement was carried out in ref [23] using 2D-TLC analysis. We clarified this point in the text.

- The vast majority of strains in the paper are locked into the low-density QS state or isolated from QS regulation. Does the high-density state come into play late in development?

As mentioned above and in the text, the low-cell-density QS state drives matrix synthesis and is responsible for launching the key initial stage of fractal wrinkling, namely the formation of confluent cell layers and the onset of curved secondary ridges. Cells in the high-cell-density QS mode do not make pellicles, rather, high-cell-density QS promotes cell dispersal from communities.

Reviewer #4 (Remarks to the Author):

Qin and Bassler investigated the structural components driving pattern formation in the pellicles of *Vibrio cholerae*. They find that matrix proteins and a quorum-sensing regulator drive the formation of the fractal patterns in pellicles. In particular, the proteins RbmC and Bap1 allow the formation of stable pellicles at the fluid interface, while the protein RbmA controls the formation of the fractal patterning. Furthermore, the expression level of RbmA controls the wavelength of the wrinkles and, when expressed at a level higher than the one normally present in the used strain, suppresses wrinkling. Lastly, the role of quorum sensing is elucidated, and two opposite mechanisms are identified to trigger and stop the patterns' formation.

The paper is well written, and the quality of the images is appropriate for publication in Nature Communications. Furthermore, the data strongly supports the conclusions of this paper on the structural components and the molecular mechanisms controlling pattern formation. Still, the impact of the pellicles composition on their material properties and, consequently, on the wrinkle formation needs to be elucidated before publication.

We thank the Reviewer for the positive comments about the writing and the data.

1. In the PNAS, 2021, the same authors develop a mathematical model to relate the pattern's emergent wavelength to its bending stiffness of the film. In the current work, what is the impact of the RbmC, Bap1, and RbmA proteins on the film mechanics? In the case of the RbmA protein, the differences in mechanical properties induced by different expression levels should also be probed. Given the amount of data already presented by the authors, I am not asking for an in-depth investigation. However, some trends in the relation between film composition/mechanics/fractal wrinkles should be investigated for completeness.

We refer the Reviewer to our previous work in which we characterized the material properties of *V. cholerae* biofilms and the contributions of RbmC, Bap1, and RbmA (Yan *et al*, Adv. Mater. 30.46 (2018): 1804153.). In that earlier work, material properties such as storage modulus, loss modulus, yield strain, and yield stress were measured using a stress-controlled parallel-plate rheometer. We reported that RbmA is responsible for endowing the biofilm with high elasticity, the VPS is crucial for providing the large strain the biofilm can support before yielding, and RbmC and Bap1, together, modulate the biofilm surface chemistry. Indeed, these material features are in line with our current observations. To ensure that readers know that the material properties have been characterized, in the revised manuscript, we have added the following text regarding biofilm mechanics/material properties in page 5 line 36:

“Previous characterization of the material properties of *V. cholerae* biofilms showed that RbmA is responsible for maintaining biofilm elasticity, VPS is crucial for the biofilm to

support large strain before yielding, and RbmC and Bap1, together, modulate the biofilm surface chemistry²⁸. Indeed, these material features are in line with our current observations.”

2. How does fractal wrinkling affect biomass accumulation and pellicles thickness? Could the thickness of the pellicles be quantified and compared between the different compositions studied?

The fractal morphology, which increases total surface area and nutrient exposure in mature pellicles, presumably confers growth advantages in dynamic environments, including those with fluid flows carrying fresh nutrients. In our current setup, however, total nutrient is fixed in the chamber at the start of the experiment for all mutants. Moreover, nutrient that remains after the pellicle reaches its mature morphology is quite limited. Therefore, we do not expect strong differences in final biomasses among our pellicles. An immediate future goal of ours is to explore dynamic environments and the influence of fractal wrinkling on community development after maturation. We hope the Reviewer can understand that measurements of pellicle thickness for all of the matrix mutants and the QS mutants studied here requires a substantial experimental undertaking that is beyond the scope of the present work. A practical constraint is that our current device is not equipped to acquire the needed data in a high throughput manner. Going forward, we plan to adapt a high-resolution imaging microscope to make it capable of high-throughput quantitation of pellicle thickness, mechanical rheology, and material properties, as dictated by matrix and QS components.

3. A minor comment regarding the style: while molecular and gene expression mechanisms are described in detail, a consistent part of the quantification of the physical parameter describing the fractal pattern is confined in the figure caption. I would suggest adding some more details about these results in the text.

We thank the Reviewer for the suggestion. In the revised manuscript, we have included passages describing the physical parameters and morphological quantitation in the text. For example, we write at page 4 line 44:

“A fractal scaling large than 1 indicates increased complexity in the surface geometry of the wrinkle features.”

And at page 6 line 1:

“...the fractal dimensions of the wrinkling features also increased (Fig. 3i), indicating the emergence of self-similar wrinkles and an increase in their geometric complexity. Additionally, the power spectra density (PSD) of the wrinkling features shifted toward smaller wavelengths (Fig. 3j), confirming the emergence of small-sized wrinkling features.”

And at page 6 line 29:

“Together, these results show that overexpression of *rbmA* generates small size wrinkles and eliminates the self-similar cascade of wrinkles that span small to large wavelengths.”

Reviewer #2 (Remarks to the Author):

Thank you for addressing all my comments and performing experiments suggested. No further comments.

Reviewer #3 (Remarks to the Author):

We find that our comments are addressed to our satisfaction. The additional data collected in response to Reviewer 2's remark about interaction between Qrr and VpsR RNAs makes the paper even stronger. The manuscript is a valuable contribution to the biofilm field and to biophysics.

Reviewer #4 (Remarks to the Author):

The Reviewers addressed all my comments, and I can now recommend the paper for publication in Nature Communications.

A minor comment: the discussion regarding biofilm thickness and mass accumulation concerning my second comment should be added to the manuscript.

Reviewer #2 (Remarks to the Author):

Thank you for addressing all my comments and performing experiments suggested. No further comments.

We thank the Reviewer for helping us improve the manuscript.

Reviewer #3 (Remarks to the Author):

We find that our comments are addressed to our satisfaction. The additional data collected in response to Reviewer 2's remark about interaction between Qrr and VpsR RNAs makes the paper even stronger. The manuscript is a valuable contribution to the biofilm field and to biophysics.

We thank the Reviewer for helping us improve the manuscript.

Reviewer #4 (Remarks to the Author):

The Reviewers addressed all my comments, and I can now recommend the paper for publication in Nature Communications. A minor comment: the discussion regarding biofilm thickness and mass accumulation concerning my second comment should be added to the manuscript.

We thank the Reviewer for the suggestion. To address this important point, we have included the following paragraph in the Discussion:

“We note that in our current setup, total nutrient is fixed in the chamber at the start of the experiment and only limited nutrient remains after the pellicle achieves its mature morphology. Therefore, we do not expect strong differences in final biomasses among the mutants. An immediate future goal is to explore dynamic environments and the influence of fractal wrinkling on community development after maturation.”